# EPIC: Efficient Personalized Index Construction for Retrieval-Augmented Generation

## Abstract

Personalizing retrieval-augmented generation (RAG) is a promising path for personal AI assistants but faces two key challenges: (i) indiscriminate indexing of large corpora imposes prohibitive memory costs, and (ii) preference-agnostic retrieval leads to mismatches with user preferences. We propose **EPIC** (Efficient Personalized Index Construction), a two-component framework that integrates preferences into both indexing and retrieval. EPIC performs *preference-aware memory refinement*, combining coarse-to-fine filtering and rewriting to build compact, preference-relevant memories, and *preference-guided embedding steering*, which adjusts query embeddings toward preference-aligned directions to improve retrieval fidelity. To benchmark personalization, we introduce three datasets—PrefWiki, PrefRQ, and PrefELI5—covering diverse domains and preferences. Experiments show that EPIC achieves the highest accuracy, the smallest memory footprint, and the lowest retrieval latency across baselines. Compared with the best-performing baseline, HippoRAG 2, EPIC improves accuracy by **10.1%p**, reduces indexing memory by **1110×**, and decreases latency by **110×**. As a plug-and-play module requiring no fine-tuning, EPIC enables efficient construction of preference-aligned memories for practical personalized RAG.

## 1 Introduction

Retrieval-Augmented Generation (RAG) has become a widely adopted paradigm for LLMs, as it grounds responses in external knowledge without requiring model retraining (Lewis et al., 2020; Lee et al., 2019; Karpukhin et al., 2020; Ram et al., 2023). By retrieving query-relevant documents at inference time, RAG enables LLMs to access up-to-date information, adapt flexibly to new domains, and produce outputs that are verifiable against trusted sources. These properties make RAG especially promising for personalized AI assistants, where outputs must align with individual user needs while operating under practical memory and compute constraints. As LLMs are increasingly deployed and embedded in everyday workflows, efficient and preference-aware retrieval is a key step toward assistants that can evolve with users over time.

Despite this potential, two critical limitations remain when applying RAG to real-world personalization (Figure 1). First, *indiscriminate indexing of massive corpora leads to prohibitive memory costs*. Typical RAG pipelines attempt to index all available documents to optimize retrieval for tasks such as single-hop, multi-hop, or long-context question answering (Sarthi et al., 2024; Jimenez et al., 2024; Gutiérrez et al., 2025). However, in real-world personalized settings, this naïve strategy becomes impractical: as a typical user may interact with thousands of new data items daily (Cai et al., 2025), indexing such ever-growing and noisy streams quickly consumes gigabytes of memory, overwhelming user devices with constrained resources. Second, *user-agnostic retrieval leads to preference mismatch*. Conventional retrievers rank documents only by semantic similarity to the query, often overlooking or even contradicting user preferences. As a result, responses may be factually correct yet misaligned with the user's intent, undermining trust in personalized assistants.

To address these challenges, we present **EPIC** (Efficient Personalized Index Construction), a framework that provides a preference-aligned RAG pipeline. EPIC integrates user preferences into both indexing and retrieval through two novel components: (i) *preference-aware memory refinement*, which performs coarse-to-fine filtering and preference-aware rewriting to reduce the memory size of a large external corpus and to highlight preference-relevant content before it is indexed; and (ii)

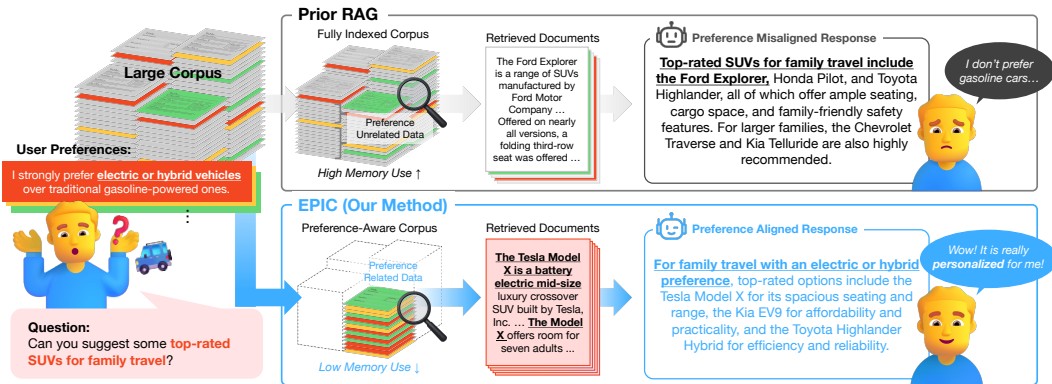

Figure 1: Comparison between previous RAG frameworks and our proposed EPIC. The figure illustrates (i) the limitations of conventional RAG pipelines, including high memory usage and preference mismatch in responses due to user-agnostic indexing, and (ii) our proposed solution, which integrates preference-based memory refinement and document retrieval to achieve memory efficiency and preference-aligned responses.

*preference-guided embedding steering*, which adjusts query embeddings toward preference-aligned directions so that preference signals are encoded directly at the embedding level, improving retrieval alignment to preference. Together, these components directly mitigate memory cost and preference mismatch, enabling the construction of compact, preference-aligned memories and ensuring generated responses are aligned to user preferences.

Preference-aware RAG lacks standard benchmarks for evaluating whether generated outputs truly reflect user preferences. Existing datasets, such as PersonaChat (Zhang et al., 2018) and PrefEval (Zhao et al., 2025), focus on narrow dialogue settings and are insufficient for studying large-scale retrieval personalization. To address this gap, we construct three new datasets: PrefWiki, with 6.9M documents, 57 personas, 570 preferences, and 2,850 questions based on the Wikipedia corpus; PrefRQ, with 6.9M documents, 90 personas, 900 preferences, and 900 questions derived from Researchy Questions (Rosset et al., 2025); and PrefELI5, with 16.4M documents, 73 personas, 730 preferences, and 730 questions based on long-form questions from ELI5 (Fan et al., 2019). Together, these datasets provide a diverse and realistic testbed for evaluating preference-aware RAG.

To measure the impact of EPIC on accuracy, memory consumption, and retrieval latency, we conducted experiments on our three preference-aware RAG benchmarks, comparing against a diverse set of baselines including three standard RAG methods, a personalized retrieval method, and three LLM-augmented indexing frameworks. Our results show that compared with the best-performing baseline, HippoRAG 2 (Gutiérrez et al., 2025), EPIC improves accuracy by **10.1%p**, reduces indexing memory consumption by **1110×**, and decreases retrieval latency by **110×**. Additionally, EPIC can be integrated with existing RAG frameworks without additional model modifications, which improves the accuracy and memory efficiency when combined. In summary, EPIC constructs preference-aligned memories while maintaining high accuracy and efficiency for personalized RAG.

## 2  RELATED WORK

### 2.1  RETRIEVAL-AUGMENTED GENERATION

Retrieval-augmented generation (RAG) has been widely studied for reducing the knowledge gap and providing more referenced information to enhance answer generation (Lee et al., 2019; Lewis et al., 2020; Gao et al., 2023). Over time, retrieval methods have advanced from traditional sparse approaches such as BM25 (Robertson et al., 2009) to dense neural retrievers like DPR (Karpukhin et al., 2020) and Contriever (Izacard et al., 2021), and recently to large-scale embedding models such as NV-Embed (Lee et al., 2025). Beyond improving the retriever itself, recent work focuses on better indexing—organizing and structuring external knowledge to improve retrieval effectiveness in RAG. RAPTOR (Sarthi et al., 2024) introduces hierarchical clustering and summarization for long context

or multi-hop reasoning, while graph-based approaches such as HippoRAG (Jimenez et al., 2024) and HippoRAG 2 (Gutiérrez et al., 2025) propagate relevance across document-entity graphs to improve multi-document reasoning. These advanced methods achieve high accuracy in answering factoid questions, but show limitations in tasks requiring responses aligned with user preferences, such as recommendation, debate, or explanation; therefore, the use of personalized RAG has emerged. Yet, personalized RAG introduces two challenges that these methods overlook. First, memory scalability, since real-world data is noisy, ever-growing, and costly to index, especially on personal devices. Second, retrieval mismatch, where documents ranked by query similarity may conflict with user preferences. We address both by curating external knowledge into compact, preference-aligned corpora, enabling RAG to produce responses that are both accurate and personalized.

## 2.2 RAG FOR PERSONALIZATION

Most existing personalized RAG systems have focused on user-authored data: they build memory for RAG from user authored documents, profiles, preference logs, or interaction histories and retrieve persona-consistent context at inference. Representative examples include PGraphRAG (Au et al., 2025), which conditions retrieval on user documents and interaction traces; EMG-RAG (Wang et al., 2024), which consolidates device-derived "personal memories" into an editable graph for downstream assistance; and PEARL (Mysore et al., 2024), which selects user-authored content to imprint individual style and values. While these methods are effective when rich personal data is available, they depart from the original goal of RAG, which is to leverage external knowledge for factual and up-to-date evidence. As such, they are not applicable to scenarios where personalization must be achieved by selecting and organizing relevant information from large open-domain corpora. Our work addresses this challenge by curating and refining external knowledge based on user preferences. This produces a compact, preference-aligned index that maintains factual accuracy while significantly reducing memory usage. As a result, the retriever can fetch information that is both relevant to the query and aligned to the user's preferences.

## 3 METHOD

**Challenges of personalized RAG.**  Realizing personalized RAG settings introduces two fundamental challenges. First, *memory scalability* becomes a critical bottleneck. Unlike benchmark settings with clean, fixed corpora, real-world personalization involves a continuous stream of evolving data such as documents, notes, and interaction history. This data is often noisy and redundant, and indexing everything without filtering quickly exhausts available memory. The problem is especially acute on personal devices, where the shift toward on-device AI requires more efficient use of memory and storage. Second, *preference misalignment* during retrieval. Standard retrievers rank documents solely based on query similarity, ignoring user-specific preferences or constraints (Zhao et al., 2025). As a result, retrieved documents may be factually accurate yet inconsistent with the user's intent. For example, a response might include correct information but conflict with the user's dietary, ethical, or stylistic preferences, reducing its usefulness and trustworthiness.

**Method overview.**  To overcome these challenges, we introduce *EPIC* (**Efficient Personalized Index Construction**) for Retrieval-Augmented Generation, illustrated in Figure 2. We depart from conventional personalized RAG systems, which dynamically retrieve user-related artifacts at inference time, our method constructs a preference-aligned knowledge base that is tailored to the user's preferences. This design is essential because effective personalization requires retrieving documents that are not only query-relevant but also preference-aligned and factually dependable at the same time. EPIC integrates user preferences into both indexing and retrieval through two key components: (i) Preference-Aware Memory Refinement (Section 3.1), which performs coarse-to-fine document filtering and rewriting to build a compact, preference-aligned external knowledge base, and (ii) Preference-Guided Embedding Steering (Section 3.2), which steers query embeddings toward related preference directions to enable preference-aware retrieval.

Together, these components enforce preference alignment at both the text and embedding levels, yielding a memory-efficient, preference-aware RAG pipeline. Moreover, EPIC is plug-and-play: it can be combined with existing RAG frameworks without modifying the underlying LLM or requiring any fine-tuning.

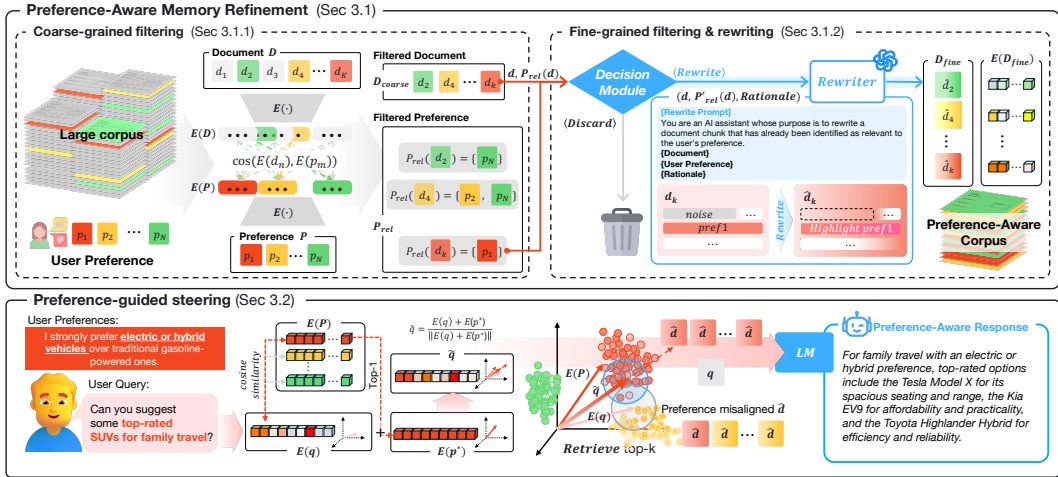

Figure 2: Overview of EPIC's pipeline. (i) *Coarse-grained filtering* (Sec. 3.1.1): documents from a large corpus are first encoded and compared with user preference embeddings; only those with at least one preference-aligned match pass this stage. (ii) *Fine-grained filtering & rewriting* (Sec. 3.1.2): the *Decision Module* verifies textual alignment and discards unrelated documents, while the *Rewriter* removes residual noise and highlights preference-relevant content to build a *preference-aware corpus*. (iii) *Preference-guided steering* (Sec. 3.2): user-query embeddings are steered toward their associated preference directions, enabling the language model to produce preference-aware responses.

### 3.1 PREFERENCE-AWARE MEMORY REFINEMENT

Indexing an entire large-scale corpus inevitably introduces two problems in personalized RAG: (i) excessive memory consumption and (ii) noisy documents that reduce retrieval quality and weaken alignment with user preferences. To address these, EPIC applies a two-stage *coarse-to-fine document filtering* pipeline that systematically reduces the corpus to a compact, preference-aligned subset.

In the first stage, coarse-grained filtering uses embedding similarity to quickly prune the corpus down to documents that are potentially relevant to user preferences while incurring only minimal computational overhead. In the second stage, fine-grained filtering uses an LM in two roles: a Decision Module that verifies genuine preference alignment, and a Rewriter that converts the retained documents into preference-aware representations. Together, these stages form an efficient, preference-aligned memory that serves as the foundation for downstream retrieval and generation.

#### 3.1.1 COARSE-GRAINED FILTERING

In this stage, we embed both documents and preferences into a shared semantic space and use cosine similarity as the relevance criterion. The computation is efficient because it involves only embedding generation and similarity scoring, without requiring additional model-based inference or reasoning. For each document, we collect the set of preferences whose similarity score exceeds a threshold. The threshold is set deliberately low so that potentially relevant documents are not excluded too early, since this step is intended only as an initial coarse filter.

Specifically, let $D$ denote the document corpus and $P = \{p_1, p_2, \ldots, p_N\}$ denote the user preference set. We embed each document and each preference into the same semantic embedding space via a sentence embedding model $\mathrm{E}(\cdot)$ (e.g., Contriever). To identify which preferences are semantically related to a document, we compute the cosine similarity between the embeddings of each document and each preference, then collect those preferences whose similarity exceeds a threshold $\tau$:

$$P_{\mathrm{rel}}(d) = \big\{p \in P \mid \cos\big(\mathrm{E}(d), \mathrm{E}(p)\big) \geq \tau\big\}, \quad \forall d \in D. \tag{1}$$

$P_{\mathrm{rel}}(d)$ denotes the subset of preferences that are semantically aligned with $d$. Based on this notion of relevance, we then restrict our attention to a candidate document set, $D_{\mathrm{coarse}}$, defined in Equation 2:

$$D_{\mathrm{coarse}} = \big\{d \in D \mid P_{\mathrm{rel}}(d) \neq \emptyset\big\}. \tag{2}$$

$D_{\mathrm{coarse}}$ collects the candidate documents with at least one aligned preference. Equations 1-2 specify the coarse-grained filtering stage, encompassing both the alignment of documents with semantically related user preferences and the construction of the candidate document set that will serve as input to the subsequent fine-grained filtering. For all $d \in D_{\mathrm{coarse}}$, $P_{\mathrm{rel}}(d)$ and $d$ are the input of the fine-grained memory refinement stage.

### 3.1.2 FINE-GRAINED FILTERING AND REWRITING

While coarse filtering efficiently prunes the search space, embedding-level similarity alone cannot capture nuanced text-level semantics. Our second stage therefore verifies preference alignment and refines the retained content. To this end, we introduce two complementary components leveraging the language understanding capabilities of an LM: a Decision Module (*DM*), which determines whether each candidate document should be discarded or retained for rewriting, and a *Rewriter*, which refines the retained documents themselves. The *DM* ensures that only documents with genuine preference relevance pass through, while the *Rewriter* removes residual noise and highlights information aligned with the matched preferences, producing text that is explicitly preference-aware before it is stored in the memory. These two modules together provide the fine-grained filtering and rewriting needed for preference-aware memory refinement, ensuring that downstream retrieval operates on text that is both compact and semantically aligned with user interests.

For each candidate document $d \in D_{\mathrm{coarse}}$ and its set of relevant preferences $P_{\mathrm{rel}}(d)$, the *DM* takes both the document and these preferences as input and produces a structured output via a rigorously defined, XML-constrained prompt (Appendix G):

$$DM(d, P_{\mathrm{rel}}) = \begin{cases} \big(\mathrm{Decision}, \mathrm{Rationale}, P'_{\mathrm{rel}}(d)\big) & \text{if Decision} = \langle\mathrm{Rewrite}\rangle \\ (\mathrm{Decision}, \mathrm{Rationale}) & \text{if Decision} = \langle\mathrm{Discard}\rangle, \end{cases} \tag{3}$$

where (i) Decision is whether Rewrite or Discard, (ii) Rationale is the explanation of the rationale of the decision, and (iii) $P'_{\mathrm{rel}}(d) \subseteq P_{\mathrm{rel}}(d)$ is the final subset of preferences directly relevant to document $d$.

For each document $d \in D_{\mathrm{coarse}}$, if the decision is $\langle\mathrm{Discard}\rangle$, it is discarded. If the decision is $\langle\mathrm{Rewrite}\rangle$, it is passed to the *Rewriter* together with its refined preference set $P'_{\mathrm{rel}}(d)$. The *Rewriter* produces a *preference-aware document*

$$\hat{d} = Rewriter\big(d, \ P'_{\mathrm{rel}}(d), \ \mathrm{Rationale}\big), \tag{4}$$

which consists of the set of all rewritten documents $D_{\mathrm{fine}}$. This *Rewriter* preserves the factual content of $d$ while filtering preference-irrelevant parts, minimizing redundancies, and highlighting preference-salient information (Appendix C for an example and Appendix H for the prompt). Each $\hat{d} \in D_{\mathrm{fine}}$ is then re-embedded to obtain its new vector representation $\mathrm{E}(\hat{d})$, which is stored in the index. By indexing these rewritten documents rather than the original text, the final index encodes user-specific signals and improves both retrieval precision and the contextual relevance of downstream generation.

This rewriting process considers preference-aware indexing and generation: by rewriting documents before indexing, user-specific signals are embedded directly at the text level, allowing the model to generate responses that better reflect user preferences when the documents are later retrieved. This also reduces the need for complex preference reasoning during generation, yielding a more computationally efficient end-to-end RAG pipeline.

### 3.2 PREFERENCE-GUIDED EMBEDDING STEERING

Although the rewriting step produces preference-aligned text, the standard embedding similarity during the retrieval phase may still fail to capture subtle preference signals in the vector space. To address this gap, EPIC *steers* user query embeddings toward the direction of their relevant preference embeddings during retrieval, so that preference alignment is enforced not only at the text level (via rewriting) but also directly at the embedding level. Notably, while documents are re-embedded after preference-aware rewriting and thus already encode preference information, the original user query embedding does not inherently capture such signals. To mitigate this asymmetry, we steer

query embeddings along preference directions, enabling them to reflect the same alignment present in the document embeddings. As a result, documents and queries that share aligned preferences are positioned in adjacent regions of the embedding space. Consequently, FAISS (Facebook AI Similarity Search) (Johnson et al., 2019) kNN retrieval operates over a reduced effective search region, thereby improving preference awareness while simultaneously reducing retrieval latency.

At query time, we first encode the user query $q$ into an embedding $E(q)$ and compare it with the set of preference embeddings. The preference $p^*$ is the preference with the highest cosine similarity with $E(q)$, and the query embedding is steered toward this preference direction:

$$\tilde{q} = \frac{E(q) + E(p^*)}{\| E(q) + E(p^*) \|}. \tag{5}$$

The steered vector $\tilde{q}$ guides the retriever to favor documents that are not only semantically related to the query but also consistent with the user's preferences, improving the relevance of the retrieved context. We assume that each user query is associated with at least one known preference; this mild assumption avoids the cold-start issue and reflects our target use cases where user profiles or preference vectors are available in advance (e.g., personalized assistants or long-term user modeling).

## 4 PREFERENCE-AWARE RAG DATASET CONSTRUCTION

**Limitations of existing datasets.** A central obstacle to advancing personalized retrieval-augmented generation (RAG) for large language models (LLMs) is the absence of standardized evaluation datasets that assess whether RAG-generated responses respect users' explicit preferences, such as "I love cats." While PrefEval (Zhao et al., 2025) is the first dataset to incorporate user's explicit preferences into language model evaluation, it is not directly suitable for RAG setups that retrieve from external knowledge sources like Wikipedia, as its preference-question pairs are derived from long conversations rather than grounded in standalone factual corpora.

**Dataset construction.** To fill this gap, we construct three new datasets: *PrefWiki*, *PrefRQ*, and *PrefELI5*, each designed for evaluating preference-aware RAG. Building on the prompt design of PrefEval, we adapt and extend its methodology to generate preference-question pairs grounded in external knowledge. Table 1 summarizes the datasets with examples. PrefWiki samples preferences from PrefEval and generates corresponding questions that can be answered using English Wikipedia (Wikimedia Foundation, 2025). PrefRQ and PrefELI5 start from existing questions in Researchy Questions (Rosset et al., 2025) and ELI5 (Fan et al., 2019), respectively, and generate matched user preferences. All pairs are validated using a language model to ensure that the preference meaningfully aligns with or conflicts with the question. Since each question is grounded in an external corpus, these datasets are directly compatible with RAG setups that retrieve and reason over factual knowledge. Due to the memory constraints of other LLM-augmented baselines, we use 10,000 randomly sampled documents of each corpus for evaluation unless stated otherwise. Further details of our dataset are provided in Appendix E. We will release the datasets upon publication.

**PrefWiki.** PrefWiki focuses on generating suggestion-style questions grounded in Wikipedia that reflect user likes or dislikes. Starting from 1,000 preferences in PrefEval, we use GPT-o3 to retain 583 that can be reasonably grounded in Wikipedia. For each preference, we generate five questions using modified PrefEval prompts, resulting in 2,915 preference-question pairs suitable for factual, personalized exploration. These pairs are later used in persona construction (Appendix E.3).

**PrefRQ.** PrefRQ targets reasoning-centered questions that reflect subjective values in domains such as philosophy, ethics, and politics. We select 1,269 questions from Researchy Questions with the highest subjectivity scores and generate aligned preferences using adapted PrefEval prompts. After LLM-based validation (Appendix E.2), 1,077 valid preference-question pairs are retained for persona construction.

**PrefELI5.** PrefELI5 focuses on aligning user preferences with explanatory questions from the ELI5 dataset. We randomly sample 10,000 questions with complete metadata and generate one preference per question using the same prompting schema. After LLM-based filtering (Appendix E.2), 3,888 pairs are retained, from which 734 high-confidence pairs are selected through manual curation for persona construction.

Table 1: Summary of our constructed preference-aware RAG benchmarks with example preference–question pairs.

| Dataset | Task | Corpus (# Documents) | # Personas | # Preferences | # Questions |
|---|---|---|---|---|---|
| **PrefWiki** | Recommendation | Wikipedia (6.9M) | 57 | 570 | 2,850 |

> Preference: *I dislike games with excessive backtracking or repetitive level design.*
> Question: *What are some of the best classic adventure games I should try?*

| | | | | | |
|---|---|---|---|---|---|
| **PrefRQ** | Debate | Wikipedia (6.9M) | 90 | 900 | 900 |

> Preference: *I prioritize economic outcomes over historical or ethical considerations when evaluating historical events.*
> Question: *Was the British Empire a force for good or bad?*

| | | | | | |
|---|---|---|---|---|---|
| **PrefELI** | Explanation | Common Crawl (16.4M) | 73 | 730 | 730 |

> Preference: *I prefer insights that highlight cultural and historical factors over economic or business strategy explanations.*
> Question: *Why are stores and restaurants on the east and west coasts of the US so different?*

## 5 EXPERIMENTS

### 5.1 SETUP

**Baselines.** To assess the effectiveness of our preference-aware, compact index construction, we compare **EPIC** with a range of indexing-focused RAG baselines, from traditional methods to state-of-the-art LLM-augmented frameworks. First, we consider a standard RAG pipeline that simply encodes every document into embeddings and retrieves the top-$k$ most similar chunks using FAISS (Facebook AI Similarity Search) (Johnson et al., 2019). This naïve RAG configuration is tested with three representative retrievers: the classic sparse matcher **BM25** (Robertson et al., 1995; Roberts et al., 2020), the dense dual-encoder **Contriever** (Izacard et al., 2021), and the large-scale embedding model **NV-Embed** (Lee et al., 2025). We also include **Pref-Appended** (Zhou et al., 2024), a simple personalized retrieval method that rewrites each query with user preferences and derives the query embedding from the rewritten query using the prompt of Zhou et al. (2024). Next, we compare against three recent *LLM-augmented indexing frameworks* that enhance retrieval beyond simple vector similarity: **RAPTOR** (Sarthi et al., 2024), which performs hierarchical clustering and summarization to improve long-document and multi-hop reasoning; **HippoRAG** (Jimenez et al., 2024); and **HippoRAG 2** (Gutiérrez et al., 2025), both of which exploit document–entity graphs to propagate relevance signals and strengthen multi-document reasoning. Finally, to demonstrate the plug-and-play nature of EPIC, we combine our preference-aware memory refinement (Section 3.1) with the strongest baseline, HippoRAG 2. Specifically, we apply our coarse-to-fine filtering and rewriting to build a preference-aligned index, and then run HippoRAG 2 on this refined corpus (**EPIC + HippoRAG 2**). This variant highlights that EPIC can be integrated with existing RAG frameworks to further improve preference-aware RAG. For RAPTOR, HippoRAG, HippoRAG 2, and EPIC, we use Contriever as their retriever. Experimental details are in Appendix D.

**Evaluation protocol and metrics.** Our evaluation centers on the quality of retrieved documents and whether they align with user preferences. To focus on the document quality, we decouple it from the generation stage. We adopt PrefEval's generation and evaluation setup across all baselines, fixing the prompts, decoding settings, and answer generator. This ensures that performance differences come *only from the retrieved content*, not from variation in how the LLM produces responses.

Because our setting does not include gold evidence documents, we follow PrefEval's evaluation protocol (Zhao et al., 2025). This protocol uses an LLM-as-judge with a structured rubric to assign four binary error labels to each response: preference-unaware, preference hallucination, inconsistency, and unhelpfulness. Preference-following accuracy is then defined as the proportion of responses with no errors under these labels. We use this measure as our primary metric and refer to it simply as *Accuracy*.

To quantify indexed documents and memory efficiency, we report *Memory Usage* in MB by serializing its full retrieval state (e.g., raw documents, vector indexes, summaries, graphs, and auxiliary

Table 2: Overall results on PrefWiki, PrefRQ, and PrefELI5 with three LLM backends. Accuracy (%) and Memory Usage (MB) are reported. Best results in each category are shown in bold.

| | Method | Llama-3.1-8B-Instruct | | | gpt-oss-20b | | | Qwen3-4B-Instruct-2507 | | |
| --- | --- | --- | --- | --- | --- | --- | --- | --- | --- | --- |
| | | PrefWiki | PrefRQ | PrefELI5 | PrefWiki | PrefRQ | PrefELI5 | PrefWiki | PrefRQ | PrefELI5 |
| Accuracy (%) | BM25 (Roberts et al., 2020) | 8.77 | 29.56 | 39.59 | 16.64 | 65.00 | 47.21 | 34.81 | 51.56 | 76.99 |
| | Contriever (Izacard et al., 2021) | 14.74 | 43.33 | 53.29 | 17.99 | 65.22 | 47.49 | 31.05 | 66.7 | 82.47 |
| | NV-Embed-v2 (Lee et al., 2025) | 19.33 | 48.89 | 55.89 | 17.68 | 66.33 | 45.62 | 34.32 | 65.3 | 81.64 |
| | Pref-Appended (Zhou et al., 2024) | 12.32 | 45.67 | 52.33 | 18.19 | 65.22 | 48.10 | 32.14 | 63.22 | 78.96 |
| | RAPTOR (Sarthi et al., 2024) | 17.68 | 44.44 | 47.53 | 16.39 | 65.22 | 47.53 | 37.47 | 54.22 | 75.89 |
| | HippoRAG (Jimenez et al., 2024) | 5.51 | 11.56 | 40.96 | 16.48 | 68.00 | 47.47 | 38.84 | 30.89 | 76.99 |
| | HippoRAG 2 (Gutiérrez et al., 2025) | 20.18 | 45.22 | 60.96 | 17.75 | 63.56 | 49.45 | 33.86 | 70.11 | 81.37 |
| | **EPIC** | **24.63** | **54.67** | **65.07** | **35.04** | **73.78** | **50.02** | **59.47** | **80.67** | **90.41** |
| | **EPIC** + HippoRAG 2 | **26.98** | **63.33** | **66.85** | 30.28 | 70.11 | 48.22 | **63.89** | **86.11** | 85.07 |
| Memory Usage (MB) | BM25 (Roberts et al., 2020) | 25.21 | 25.21 | 22.84 | 25.21 | 25.21 | 22.84 | 25.21 | 25.21 | 22.84 |
| | Contriever (Izacard et al., 2021) | 142.16 | 142.16 | 133.54 | 142.16 | 142.16 | 133.54 | 142.16 | 142.16 | 133.54 |
| | NV-Embed-v2 (Lee et al., 2025) | 648.96 | 648.96 | 613.24 | 648.96 | 648.96 | 613.24 | 648.96 | 648.96 | 613.24 |
| | Pref-Appended (Zhou et al., 2024) | 142.16 | 142.16 | 133.54 | 142.16 | 142.16 | 133.54 | 142.16 | 142.16 | 133.54 |
| | RAPTOR (Sarthi et al., 2024) | 297.05 | 297.05 | 291.04 | 319.27 | 319.27 | 312.03 | 298.72 | 298.72 | 290.93 |
| | HippoRAG (Jimenez et al., 2024) | 2347.46 | 2347.46 | 2863.45 | 2385.71 | 2385.71 | 2392.44 | 2385.62 | 2385.62 | 2404.99 |
| | HippoRAG 2 (Gutiérrez et al., 2025) | 2896.50 | 2896.50 | 2554.56 | 2815.56 | 2815.56 | 2146.95 | 2831.08 | 2831.08 | 2595.29 |
| | **EPIC** | **0.45** | **1.54** | **9.70** | **0.22** | **0.63** | **1.11** | **0.75** | **1.73** | **5.82** |
| | **EPIC** + HippoRAG 2 | 8.92 | 26.81 | 144.50 | 5.26 | 14.30 | 19.48 | 12.70 | 29.13 | 84.73 |

Table 3: Incremental ablation of EPIC components. C: Coarse-grained filtering (Sec. 3.1.1); F: Fine-grained filtering & rewriting (Sec. 3.1.2); S: Preference-guided embedding steering (Sec. 3.2). Accuracy (%) and memory usage (MB) are reported on all three benchmarks.

| | | | PrefWiki | | PrefRQ | | PrefELI5 | |
| --- | --- | --- | --- | --- | --- | --- | --- | --- |
| C | F | S | Accuracy (%) | Memory Usage (MB) | Accuracy (%) | Memory Usage (MB) | Accuracy (%) | Memory Usage (MB) |
| ✗ | ✗ | ✗ | 14.74 | 142.16 | 43.33 | 142.16 | 31.05 | 133.54 |
| ✓ | ✗ | ✗ | 10.7 | 1.83 | 42.56 | 3.28 | 50.55 | 25.93 |
| ✓ | ✓ | ✗ | 20.14 | **0.45** | 52.56 | **1.54** | 58.22 | **9.70** |
| ✓ | ✓ | ✓ | **24.63** | **0.45** | **54.67** | **1.54** | **65.07** | **9.70** |

stores). All numbers in the main text are actual on-disk MB. Because counted components differ by method, the exact inclusion rules are detailed in the Appendix D.7.

## 5.2 RESULTS

**Overall Results.** Table 2 summarizes overall results on our three datasets (PrefWiki, PrefRQ, and PrefELI5) with three LLM backends: Llama-3.1-8B-Instruct (AI@Meta, 2024), gpt-oss-20b (OpenAI, 2025), and Qwen3-4B-Instruct-2507 (Qwen Team, 2025). Across all datasets and LLMs, EPIC demonstrates the highest accuracy while simultaneously delivering orders-of-magnitude smaller memory usage than both standard RAG (BM25, Contriever, NV-Embed), a simple personalized retrieval baseline (Pref-Appended), and LLM-augmented indexing frameworks (RAPTOR, HippoRAG, HippoRAG 2). For instance, compared to the state-of-the-art RAG framework (HippoRAG 2), EPIC showed 10.1%p accuracy improvement and $1110\times$ less memory usage on average. We attribute these gains to (i) the removal of preference-irrelevant noise via coarse-grained filtering, (ii) preservation and amplification of preference-relevant content via fine-grained filtering & rewriting, and (iii) the reuse of rewritten signals for steering at both indexing and retrieval time. The hybrid method (EPIC + HippoRAG 2) further shows that EPIC can be integrated into the state-of-the-art RAG framework with powerful graph-based reasoning, achieving higher accuracy (+10.9%p) and greater memory efficiency ($70\times$) on average.

**Ablation study.** We conduct an ablation study to examine how each module of EPIC contributes to overall performance. Table 3 reports accuracy and index-memory usage across the three preference-aware benchmarks as the modules are added sequentially. Applying coarse-grained filtering alone

Table 4: Indexing efficiency of our two-stage filtering on PrefWiki using Llama-3.1-8B-Instruct as the backend, averaged over personas 0-9.

| Method | Indexing Time (s) | | | | # of Kept Chunks | Accuracy (%) |
|---|---|---|---|---|---|---|
| | Coarse Filtering | Fine Filtering | Rewriting | Index Constructing | Total | | |
| No filtering | 0 | 0 | 3698.48 | 126.53 | 3825.01 | 39920 | 10.0 |
| Coarse-only | 0.033 | 0 | 35.82 | 1.28 | 37.14 | 525 | 10.40 |
| Fine-only | 0 | 3602.59 | 402.13 | 15.86 | 4020.58 | 4507 | 13.80 |
| Coarse + Fine | 0.034 | 30.37 | 10.47 | 0.63 | 41.50 | 143 | 29.6 |

reduces memory consumption by an average of $13.46\times$, showing that early embedding-level filtering effectively reduces the volume of irrelevant documents. Adding fine-grained filtering & rewriting increases accuracy across all benchmarks and compresses the index by an additional $2.65\times$, since rewriting produces a more compact, preference-aligned corpus. Integrating preference-guided embedding steering shows the largest accuracy gains while leaving memory usage unchanged. These results demonstrate that coarse filtering delivers the main memory savings, fine-grained rewriting strengthens preference alignment and further reduces memory size, and embedding steering leverages the refined preference cues at the embedding level to maximize retrieval accuracy.

**Indexing efficiency of our two-stage filtering.**   We further analyze our two-stage filtering strategy to highlight the synergy between coarse-grained and fine-grained filtering, showing why both are necessary. Table 4 summarizes the results on the PrefWiki benchmark, including the time taken for each stage, the number of chunks retained, and final retrieval accuracy. Coarse-only reduces the index from nearly 40,000 chunks to about 500 in under 40 seconds, showing the fastest indexing time but with a slight drop in accuracy. Fine-only requires more than an hour of LLM calls and still leaves over four thousand chunks, offering better accuracy than no filtering, but at a prohibitive time cost. Combining the two stages achieves the best trade-off; total indexing time falls to roughly 40 seconds, the index shrinks to about 140 chunks, and accuracy rises to 26%, surpassing all single-stage variants. This demonstrates that fast coarse filtering sharply narrows the candidate set, enabling fine-grained filtering and rewriting to focus on a small, high-quality subset and ultimately bring both higher accuracy and lower preprocessing cost.

**Comparison of retrieval latency.**   Figure 3 shows a comparison of retrieval latency across different methods. As shown, multi-stage retrievers (RAPTOR, HippoRAG, and HippoRAG 2) incur substantially higher latency (at least 300 ms) than single-pass dense/sparse retrievers due to multiple k-nearest neighbor (kNN) probes, tree/graph traversals, and re-ranking. EPIC preserves Contriever's single flat-index kNN; although our preference-guided embedding steering introduces a slight per-query overhead, the significantly smaller index shrinks the candidate set and scan, resulting in a net reduction in end-to-end retrieval time–even achieving lower latency (3.5 to 4 ms) than Contriever (around 6.4 ms).

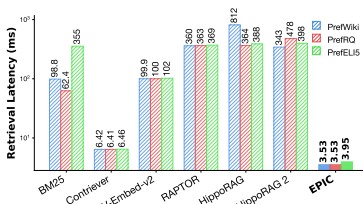

Figure 3: Average end-to-end per-query retrieval latency on the three preference-aware benchmarks.

## 6 CONCLUSION

We introduced EPIC, a framework for Efficient Personalized Index Construction that builds a memory-efficient and preference-aligned knowledge base for RAG. EPIC integrates (i) preference-aware memory refinement, which performs coarse-to-fine filtering and rewriting to remove preference-irrelevant noise and compress the corpus, and (ii) preference-guided embedding steering, which aligns both document and query embeddings with user preferences to improve retrieval accuracy. Across three constructed preference-aware RAG benchmarks—PrefWiki, PrefRQ, and PrefELI5—EPIC consistently outperforms strong RAG baselines. Looking forward, we see EPIC as a stepping stone for next-generation personalized AI assistants that must adapt to evolving user needs under tight resource constraints.

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

## A  LIMITATION

Our current design assumes a single user's explicit preferences and evaluates efficiency primarily in static, offline corpora. Future work should examine how the method scales to multi-user environments or to continually updated streams, where preference drift and dynamic data injection could challenge both memory efficiency and retrieval alignment. Because preference alignment necessarily depends on explicit user input, biased or harmful preferences might propagate into the indexed corpus; careful governance and transparent preference collection will be essential when applying EPIC in real-world personalized assistants.

## B  DETAILED RELATED WORK

### B.1  PARAMETRIC KNOWLEDGE-BASED PERSONALIZATION

Mainstream alignment via instruction tuning and reinforcement learning from human feedback (RLHF) optimizes for aggregate human preferences (Ouyang et al., 2022), which can dilute or even conflict with the idiosyncratic needs of specific users. Early personalization approaches adapt model parameters through full fine-tuning or alignment with user-specific data, but these approaches are computationally and operationally expensive at scale (e.g., separate fine-tuned checkpoints per user). Parameter-efficient fine-tuning (PEFT) mitigates these costs by updating small subsets or low-rank projections while freezing most weights. Adapters (Houlsby et al., 2019) insert lightweight modules between transformer layers; LoRA (Hu et al., 2022) injects trainable low-rank matrices and reports up to 10,000× fewer trainable parameters and 3× lower memory versus full fine-tuning on GPT-3-class models; prefix-tuning (Li & Liang, 2021) and prompt-tuning (Lester et al., 2021) learn continuous "soft prompts" that steer frozen backbones. While effective, these techniques still require per-user artifacts (raising storage, routing, and lifecycle overhead), can suffer from forgetting or entanglement under continual updates, and may pose privacy concerns when centralizing user-specific gradients or weights. Recent work explores explicitly personalized PEFT and privacy-aware variants. One PEFT Per User (OPPU) (Tan et al., 2024) attaches user-specific PEFT modules that can be plugged into a shared base model and combined with non-parametric user profiles, improving personalization while preserving model ownership and reducing central data exposure.

### B.2  USER PREFERENCE DATASET

User Preference Datasets collect examples in which a user's tastes, constraints, or style (e.g., likes/dislikes, tone, format, accessibility needs) are stated or implied, and models are evaluated on whether they respect those preferences in their responses. Early work centered on persona-conditioned dialogue (e.g., profile sentences guiding open-domain conversation). More recent researches include longer-context interactions where a model must infer, remember, and apply preferences over multiple turns. While valuable for personalized generation, most such datasets were not designed to directly test retrieval over an external corpus. Personalized RAG must retrieve documents that satisfy both the information need query and the user's preferences, then ground the answer on those documents. Existing user-preference datasets rarely support rigorous evaluation of this retrieval objective for several reasons:

1. Either the user preference or the question is missing, so the retrieval target cannot be precisely defined.

2. Questions rarely induce preference conflicts, making violations unlikely and the retrieval task non-discriminative.

3. No gold labels tying (preference, question) pairs to documents that both answer the query and satisfy preferences.

In light of these limitations of existing datasets, this study makes extensive use of the PrefEval benchmark (Zhao et al., 2025).

### B.3 PREFEVAL BENCHMARK

The Explicit Preference subset of PrefEval dataset (Zhao et al., 2025) focuses on preferences the user states unambiguously (e.g., "I avoid electric vehicles," "I prefer spicy food"). Instances typically pair:

1. a preference statement (clear like/dislike or constraint), and

2. a query that can easily elicit a default answer which would violate that preference unless the model takes it into account (e.g., recommending the best compact cars for city driving, where the most top options are electric vehicles),

3. optionally, a short explanation/rationale highlighting why the query is risky with respect to the preference.

This subset deliberately booby-traps the obvious answer: the quickest generic response is often preference-inconsistent. Strong performance therefore requires the model to (1) recognize the explicit constraint, (2) prioritize it alongside topical relevance, and (3) surface alternatives that respect the constraint. "The four error types are: (1) Preference-Unaware Violation: The LLM provides generic recommendations that contradict the user's stated preference due to unawareness of user preference. (2) Preference Hallucination Violation: The response fabricates or misattributes preferences, diverging from the user's true preference and violates the true preference. (3) Inconsistent Violation: The response acknowledges the correct preference but generates contradicting response. (4) Unhelpful Response: The response lacks relevant recommendations or fails to address the query due to poor recall of the user's preference. To validate our LLM-based evaluation method, we manually checked 200 randomly sampled evaluations, with an observed 5% error rate. This demonstrates strong agreement between human judgment and LLM-based assessments with Claude 3 Sonnet." Importantly, a notable aspect of PrefEval is its evaluation methodology, which leverages an LLM-based judge to categorize errors in preference following. Instead of relying solely on BLEU/ROUGE or costly human ratings, PrefEval uses an LLM to automatically check each generated response against the user's stated preference. The evaluation defines four possible error types (failure modes), aggregated from binary criteria:

1. Preference-Unaware Violation: The LLM provides generic recommendations that contradict the user's stated preference due to unawareness of user preference.

2. Preference Hallucination Violation: The response fabricates or misattributes preferences, diverging from the user's true preference and violates the true preference.

3. Inconsistent Violation: The response acknowledges the correct preference but generates contradicting response.

4. Unhelpful Response: The response lacks relevant recommendations or fails to address the query due to poor recall of the user's preference.

## C EXAMPLES OF PREFERENCE-AWARE REWRITING

---

**Example 1**

Input:

**Original Document:** *Neal Ronaghan of "Nintendo World Report" enjoyed the permadeath feature, saying "Every run is engrossing and fun and when I die, the only thought rushing through my brain is to start over and try again, pushing as far past my previous run as I can."\nSales.\nAbout a year from its early access release, "Dead Cells" sold over 730,000 units, and exceeded 850,000 units just prior to its full release. By May 2019, within ten months of its full release, "Dead Cells" had accumulated sales of two million units. In March 2021, "Dead Cells" had sold 5 million copies during the announcement of their third DLC.\nLegacy.\nIn June 2023, developer Motion Twin announced that an animated series, titled "Dead Cells: Immortalis", was in the works by French animation studio Bobbypills, the studio behind their animated trailers, and was released on June 19th, 2024 by French animation network, ADN (Animation Digital Network) and lasted for 10 episodes.*

**Relevant Preference:** *I dislike games with excessive backtracking or repetitive level design.*

**Reason:** *The given chunk is relevant to the user's preference because it discusses a game ("Dead Cells") that has a permadeath feature, which implies a level of difficulty and challenge that may be appealing to players who enjoy games with a sense of progression and improvement.*

Output:

**Rewritten Document:** *"Dead Cells" features permadeath, where every run is a new challenge, and players must push past their previous achievements. The game's success is evident in its sales, with over 5 million copies sold by March 2021, and the announcement of an animated series, "Dead Cells: Immortalis", which explores the game's world and characters.*

---

**Example 2**

Input:

**Original Document:** *Form of nonverbal communication\nEye contact occurs when two people or non-human animals look at each other's eyes at the same time. In people, eye contact is a form of nonverbal communication and can have a large influence on social behavior. Coined in the early to mid-1960s, the term came from the West to often define the act as a meaningful and important sign of confidence and respect. The customs, meaning, and significance of eye contact can vary greatly between societies, neurotypes, and religions.\nThe study of eye contact is sometimes known as "oculesics".\nSocial meanings.\nEye contact and facial expressions provide important social and emotional information.*

**Relevant Preference:** *I prefer face-to-face interactions and avoid digital communication as much as possible.*

**Reason:** *The given chunk is relevant to the user's preference for face-to-face interactions, as it discusses nonverbal communication and social behavior, which are essential aspects of in-person interactions.*

Output:

**Rewritten Document:** *Eye contact is a form of nonverbal communication that plays a significant role in social behavior, conveying confidence and respect. It provides important social and emotional information, essential for effective face-to-face interactions.*

---

# D EXPERIMENTAL DETAILS

## D.1 WIKIPEDIA CORPUS

PrefWiki and PrefRQ rely on the English Wikipedia as the underlying retrieval corpus. We use the official dump released on 2025-04-04 (`enwiki/latest`).[1] Following Chen et al. (2017), we process the dump using the `WikiExtractor` script[2], which removes MediaWiki markup and retains only plain text. Each processed article is stored in JSON format with two fields: `title` and `text`. The extracted snapshot contains 6,945,964 documents totaling 17.74 GB of plaintext. For index construction in our experiments, we uniformly sample 100,000 articles and segment them into retrieval units ("chunks"), resulting in 398,861 chunks. This chunked corpus is used consistently across PrefWiki, and PrefRQ evaluations.

## D.2 ELI5 CORPUS

From the full collection of supporting documents provided by the ELI5 dataset—comprising 16,453,150 documents (approximately 185 GB)—we randomly sample 20,000 documents. These are preprocessed and segmented into retrieval units ("chunks") following the same procedure applied to the Wikipedia corpus. This yields a total of 379,994 chunks, which is comparable in scale to the chunked Wikipedia corpus. The resulting chunked corpus is employed exclusively for the PrefELI5 evaluation.

## D.3 CHUNKING STRATEGY

In retrieval-augmented generation, source texts must be divided into smaller segments to enable precise retrieval. Without chunking, retrieval systems risk pulling in overly broad or irrelevant sections, thereby diminishing contextual alignment and response quality. In our experiments, we implemented a fixed-size chunking strategy with semantic safeguards. Source documents were first segmented into chunks of approximately 100 words. To preserve coherence, when a single sentence exceeded the 100-word threshold, we retained the entire sentence within the chunk rather than dividing it. This approach ensures that semantic integrity and contextual continuity are not compromised by arbitrary truncation. By structuring the data in this way, each chunk becomes a self-contained unit of meaning, allowing the retrieval system to assess its relevance independently. Importantly, this strategy was not only applied to our proposed method, EPIC, but also consistently enforced across all baseline models in our experiments. By adopting an identical preprocessing pipeline for every system under evaluation, we ensured that performance comparisons reflect genuine methodological differences rather than artifacts of data segmentation.

## D.4 MODELS AND INFERENCE

We evaluate three instruction-tuned LLMs of different scales: Llama-3.1-8B-Instruct, gpt-oss-20b, and Qwen3-4B-Instruct-2507. All inference is conducted using the `vLLM` engine (Kwon et al., 2023), which provides optimized memory management and high-throughput serving. For reproducibility, we fix the random seed to `0`, set the generation temperature to `0.0`, and set the computation `dtype` to `float32`. We adopt vLLM to ensure consistent serving latency across methods and to enable scalable experimentation without incurring additional memory overhead.

## D.5 RETRIEVER AND INDEXING

We encode all chunks and preferences using Contriever embeddings (dimension `768`). Vector indexing and nearest-neighbor search are implemented with FAISS (Johnson et al., 2019), using the `IndexFlatIP` backend for exact inner-product kNN search with $k = $ xx.

---

[1] https://dumps.wikimedia.org/enwiki/latest/
[2] https://github.com/attardi/wikiextractor

Table 5: Analysis on Memory Usage. PrefWiki, Llama-3.1-8B-Instruct

| | Method | Raw Documents | FAISS Index | RAPTOR Tree | HippoRAG Graph | Embeddings | Misc. | Total |
|---|---|---|---|---|---|---|---|---|
| Memory Usage (MB) | BM25 (Robertson et al., 2009) | 25.21 | 0 | 0 | 0 | 0 | 0 | 25.21 |
| | Contriever (Izacard et al., 2021) | 25.21 | 116.95 | 0 | 0 | 0 | 0 | 142.16 |
| | NV-Embed-v2 (Lee et al., 2025) | 25.21 | 623.75 | 0 | 0 | 0 | 0 | 648.96 |
| | Pref-Appended (Zhou et al., 2024) | 25.21 | 116.95 | 0 | 0 | 0 | 0 | 142.16 |
| | RAPTOR (Sarthi et al., 2024) | 0 | 0 | 297.05 | 0 | 0 | 0 | 297.05 |
| | HippoRAG (Jimenez et al., 2024) | 26.91 | 0 | 0 | 1.54 | 2282.44 | 36.57 | 2347.46 |
| | HippoRAG 2 (Gutiérrez et al., 2025) | 0 | 0 | 0 | 130.74 | 2765.76 | 0 | 2896.50 |
| | **EPIC** | 0.04 | 0.41 | 0 | 0 | 0 | 0 | 0.45 |
| | **EPIC** + HippoRAG 2 | 0 | 0 | 0 | 0.28 | 8.51 | 0.13 | 8.92 |

### D.6 ENVIRONMENTS

All experiments are run on a single server with 2 $\times$ AMD EPYC 9354 CPUs (32 cores / 64 threads each; 64 cores / 128 threads total) and 8 $\times$ NVIDIA RTX 6000 Ada GPUs (48 GB each). All baselines and our methods share the same hardware and software configuration to ensure a fair comparison.

### D.7 MEMORY USAGE MEASUREMENT

Table 5 presents the memory usage requirements (in MB) for various Retrieval-Augmented Generation (RAG) methods across different indexing strategies. The breakdown highlights the memory consumed by raw document storage, FAISS index, RAPTOR tree, HippoRAG graph structure, embeddings, and additional miscellaneous components.

# E DATASET CONSTRUCTION DETAILS

## E.1 KNOWLEDGE BASE OF DATASET

Each dataset is built upon a large-scale corpus and is annotated with personas, preferences, and corresponding questions. The English Wikipedia dump released on 2025-04-04 serves as the primary corpus, while ELI5 supporting documents are retrieved from the August 2018 Common Crawl snapshot.

## E.2 PREFERENCE-QUESTION VALIDITY ASSESSMENT

The Validity Assessment procedure is designed to ensure that each preference–question pair is suitable for evaluating preference-aware reasoning. Specifically, we consider a pair invalid under the following conditions:

1. the question directly contradicts the user's preference, such that any answer would inherently violate the preference

2. the question is already perfectly aligned with the preference, such that no additional reasoning about the preference is required

3. the question has a negligible probability of violating the preference under the PrefEval data generation prompt, i.e., when $P(\text{answer} \mid \text{question}) \ll P(\text{answer} \mid \text{preference}, \text{question})$, indicating that even without conditioning on the preference, natural answers rarely conflict with it

For PrefELI5, all three conditions are applied. For PrefRQ, since the dataset is pre-filtered to contain highly subjective questions from the Researchy Questions corpus, only conditions (1) and (2) are checked. The questions in Researchy Questions were processed such that, whenever the preference–question pair generated with the corresponding generated preference was assessed as invalid in the validity assessment, the preference was regenerated. If ten consecutive invalid results occurred, the given question was regarded as unsuitable and was excluded from use in the dataset.

---

**Prompt for Preference-Question Validity Assessment**

```
You are an expert evaluator for preference-question pairs designed to test AI systems'
ability to respect user preferences. Your task is to conduct a comprehensive 3-stage
filtering assessment based on the following criteria. Each pair must pass ALL three
criteria to be considered VALID.

1. Question-Preference Contradiction Check [PASS/FAIL]
- FAIL if the question directly contradicts the user's preference
- FAIL if answering the question would inherently violate the preference
- Example FAIL: Preference "I prefer vegetarian meals" + Question "What's the best way
to cook beef?"

2. Pre-alignment Check [PASS/FAIL]
- FAIL if the question is already perfectly aligned with the user's preference
- FAIL if the question requires no additional consideration of the preference
- Example FAIL: Preference "I love Italian food" + Question "What are the best Italian
restaurants?"

3. Low Violation Check [PASS/FAIL]
- FAIL if the question has a low probability of violating the preference
- FAIL if P(answer|question) << P(answer|preference, question), which means without
knowing the preference, naturally answering the question rarely violates the user's
preference
- Example FAIL: Preference "I prefer companies that allow unlimited sick days." +
Question "Which companies have the most generous sick leave policies?"

Provide your assessment in the following XML format:
<evaluation>
    <contradiction_check>
        <result>PASS/FAIL</result>
        <explanation>Only if FAIL</explanation>
    </contradiction_check>
```

```
        <prealignment_check>
            <result>PASS/FAIL</result>
            <explanation>Only if FAIL</explanation>
        </prealignment_check>
        <violation_check>
            <result>PASS/FAIL</result>
            <explanation>Only if FAIL</explanation>
        </violation_check>
        <final_assessment>
            <verdict>VALID/INVALID</verdict>
        </final_assessment>
    </evaluation>
```

### E.3   PERSONA CONSTRUCTION

An overview of the evaluation datasets is provided in Table 1, which highlights the distribution of personas, preferences per persona, and questions per preference across PrefWiki, PrefELI5, and PrefRQ. For all three datasets, personas are constructed via a common procedure. From the available pool, sets of ten preferences are sampled uniformly at random and checked by an LLM for internal conflicts between ten preferences. If no conflict is detected, the ten-item set is instantiated as a persona and the corresponding preferences are removed from the pool. This process repeats until the pool contains fewer than ten items or thirty consecutive conflict-free samples cannot be found. As a result, of the 583 PrefWiki preferences, 570 are utilized; in PrefRQ, 900 of 1,077 preferences are incorporated; and in PrefELI5, 730 of 734 pairs are retained.

# F DATASET EXAMPLES

## F.1 EXAMPLES OF DATA IN PREFWIKI

Table 6: Examples of PrefWiki.

| Persona Index | Preference (3 out of 10) | Question |
|---|---|---|
| Persona #0 | I dislike pickup trucks because I find them too large and impractical. | What's a popular American vehicle model I should consider for off-road adventures? |
| | | What are some of the top-selling vehicles in the United States that I should look into? |
| | | Which vehicles are known for their towing capacity that I should research? |
| | | What vehicles are highly recommended for road trips across the United States? |
| | | Can you suggest some vehicles with a strong reputation for durability and reliability? |
| | I dislike games with excessive backtracking or repetitive level design. | What are some of the best classic adventure games I should try? |
| | | Could you suggest some of the most highly acclaimed RPGs for me to play? |
| | | What are some popular Metroidvania games that I might enjoy? |
| | | Can you recommend any influential platform games I should add to my collection? |
| | | What are some must-play open-world games that offer a deep story? |
| | I follow a strict vegan diet and refuse to consume any animal-derived products, including honey. | What are some of the must-try traditional dishes when visiting Athens, Greece? |
| | | Which famous restaurants in Tokyo should I visit for an authentic Japanese dining experience? |
| | | Can you recommend iconic street foods to try while traveling in Bangkok, Thailand? |
| | | What are the best culinary experiences to have in Paris, France? |
| | | What are the classic dishes to try in New Orleans, Louisiana, for a taste of local culture? |

## F.2 EXAMPLES OF DATA IN PREFRQ

Table 7: Examples of PrefRQ.

| Persona Index | Preference | Question |
|---|---|---|
| Persona #0 | I prioritize emotional intelligence over hard skills as the key to success. | what makes a person successful? |
| | I prefer considering moral philosophy over legal or political implications when discussing justice. | is capital punishment justified |
| | I prioritize practical and utilitarian skills over artistic or literary pursuits in education. | do we really need literature in our lives |
| | I prioritize healthcare systems that are driven by market competition and private enterprise rather than government intervention. | is healthcare a right or a privilege? |
| | I prefer urban spaces to be strictly maintained for functionality and cleanliness over artistic expression. | is graffiti an act of vandalism or the creation of art? |
| | I prioritize ecological balance and species survival over individual animal welfare concerns. | do animals deserve rights? |
| | I strongly believe in preserving natural ecosystems without any form of geoengineering intervention. | how has climate change affected you |
| | I strongly prefer leaders who prioritize environmental sustainability over economic growth. | what do people look for in a president |
| | I prioritize secular reasoning over religious beliefs in decision-making processes. | should parents deny their children medical treatment because of religious beliefs |
| | I prefer advancements in environmental sustainability over economic growth in industrial sectors. | what interests you about the construction industry |
| Persona #1 | I believe that financial matters should be addressed solely through collective bargaining and union negotiations, rather than individualistic approaches. | how women feel about the gender pay gap |
| | I focus exclusively on economic and trade outcomes when evaluating historical events. | was the american revolution good or bad |
| | ⋮ | |

## F.3 EXAMPLES OF DATA IN PREFELI5

Table 8: Examples of PrefELI5.

| Persona Index | **Preference** (3 out of 10) | **Question** |
|---|---|---|
| Persona #0 | I prefer explanations grounded in mathematical proof and rigor rather than those based on aesthetics or mystical interpretations. | The Golden Ratio and how it relates to the world around us and the Fibonacci Sequence: Please |
| | I prefer cosmic phenomena explanations over subatomic particle explanations. | On a linear scale, can we s̈eefurther into outer space or inner space?: What's the smallest thing we're aware of and the largest or furthest away?Where are we on that scale? |
| | I prefer insights that highlight cultural and historical factors over economic or business strategy explanations. | Why are stores and restaurants on the east and west coasts of the US so different?: Some examples being in n out only on the west coast, Walmart barely on the west coast compared to the east, and so many other establishments. Why are they so isolated to one side? |
| Persona #1 | I strongly prefer explanations that emphasize cultural and historical perspectives over astronomical or geometric reasoning. | Why is North considered 'up'?: Why aren't maps orientated so that the northern hemisphere appears on the bottom and not vice versa? |
| | I prefer explanations that highlight the cultural significance and human achievements over aesthetic or engineering marvels. | What dictates a Wonder of the world?: I'm a bit confused as to why there are only 14, 7 from ancient and modern world, and why they chose those 7 for each specifically. For the longest time I thought stonehenge was a wonder, but it wasn't, as well as the easter island heads, those things were full of 'wonder' as people couldn't figure them out. But they aren't put as wonders. |
| | I prefer psychological explanations based on cognitive behavioral principles rather than neurological or genetic theories. | Why Do I Feel The Need To Do Something To One Side Of My Body After Doing It To The Other?: For Example : I touch my left ear, now I have the urge to touch my right one! Why is that? |

## G  PROMPTS USED FOR FINE-GRAINED FILTERING

---

**Prompt for Fine-grained Filtering**

```
<identity> You are an AI assistant whose purpose is to analyze and determine whether
the chunk is relevant to user's predefined preferences.
</identity>

<planning_steps>
1. Understand all user preferences thoroughly.
2. Read the given document chunk.
3. If the chunk contains no content relevant to any of the preferences, decide:
Discard.
4. If the chunk is relevant to any preference, or can be rewritten to align with any
stated preference, decide: Rewrite.
5. Always explain the reason clearly.
6. If Rewrite, specify exactly which preferences the chunk aligns with.
7. Output must strictly follow the XML structure and include only XML.
</planning_steps>

<guidelines>
- Do not infer unstated preferences.
- When listing <relevant_preferences>, use the exact preference texts as provided by
the user, do not paraphrase or modify.
</guidelines>

<response_requirements>
- Every output must follow strict XML format.
- The <reason> must explicitly state why the chunk should be rewritten, or discarded.
- If <decision> is Rewrite, the <relevant_preferences> tag must be present and list
the matched preferences.
- Wrap each relevant preference in its own <preference> tag within the
<relevant_preferences> section.
- <preference> tags must contain the exact preference text as originally stated by the
user; no generalization or paraphrasing.
- No extra text outside the XML is allowed.
</response_requirements>

<user_preferences>
{preference}
</user_preferences>

<given_chunk>
{chunk}
</given_chunk>

<task>
Decide whether to Discard or Rewrite the given chunk based on alignment with the
listed user preferences.

Follow these rules:
- If the chunk is unrelated, choose <decision>Discard</decision>.
- If the chunk is relevant, choose <decision>Rewrite</decision>, and include matched
<relevant_preferences>.
- Always include a <reason> explaining the decision.
- Output only a single <answer> XML block in strict XML format, with no extra
explanation or commentary.
</task>

<answer>
```

---

## H PROMPTS USED FOR REWRITING

---

**Prompt for Rewriting**

```
<identity>
You are an AI assistant whose purpose is to rewrite a document chunk that has already
been identified as relevant to the user's preference.
</identity>

<planning_steps>
1. Read the user's stated preference.
2. Read the document chunk.
3. Read the given reason for why this chunk was marked for rewriting.
4. Rewrite the chunk concisely, while:
   - Preserving the core factual content.
   - Emphasizing the parts that align with the preference.
5. Do not add commentary, reasoning, or inference beyond what is in the text.
6. Output must consist of a single <rewrite> XML tag.
</planning_steps>

<guidelines>
- You do not need to assess relevance | the chunk has already been judged relevant.
- Focus on making relevant content clearer and more concise.
- Only output the rewrite. Do not include a <reason> or any other tag.
- Never fabricate or assume additional information.
- Avoid the use of pronouns (e.g., "he," "she," "they," "it," "this," "that").
Instead, use specific nouns or rephrased structures to maintain clarity and precision.
</guidelines>

<response_requirements>
- Output must contain only a single <rewrite>...</rewrite> XML tag.
- No additional text, no explanation, and no other tags.
- The rewrite must emphasize preference-relevant content.
</response_requirements>

<user_preferences>
{preferences}
</user_preferences>

<given_chunk>
{chunk}
</given_chunk>

<reason>
{reason}
</reason>

<task>
Generate a concise, preference-aware rewrite of the given chunk.

- Write a concise rewrite using only the <rewrite> XML tag.
- Emphasize content related to the user's preference.
- Do not include any explanation or other tags.
</task>

<answer>
```

---

# I EXPERIMENT ON LARGE CORPUS

Table 9: Comparative evaluation results on PrefWiki, PrefRQ, and PrefELI5 using Llama-3.1-8B-Instruct. The left three columns report performance with a 10,000-document subset of the corpus, while the right three columns report performance with a 100,000-document subset used as the external knowledge source.

| | Method | Small Scale Corpus | | | Large Scale Corpus | | |
|---|---|---|---|---|---|---|---|
| | | PrefWiki | PrefRQ | PrefELI5 | PrefWiki | PrefRQ | PrefELI5 |
| Accuracy (%) | BM25 (Roberts et al., 2020) | 8.77 | 29.56 | 39.59 | 11.12 | 38.56 | 47.81 |
| | Contriever (Izacard et al., 2021) | 14.74 | 43.33 | 53.29 | 21.02 | 51.89 | 37.40 |
| | NV-Embed-v2 (Lee et al., 2025) | 19.33 | 48.89 | 55.89 | 25.3 | 55.56 | 59.73 |
| | HippoRAG 2 (Gutiérrez et al., 2025) | 20.18 | 45.22 | 60.96 | 25.82 | 56.00 | 61.37 |
| | EPIC | **24.63** | **54.67** | **65.07** | **40.98** | **65.33** | **73.29** |
| Memory Usage (MB) | BM25 (Roberts et al., 2020) | 25.21 | 25.21 | 22.84 | 252.44 | 252.44 | 230.21 |
| | Contriever (Izacard et al., 2021) | 142.16 | 142.16 | 133.54 | 1420.98 | 1420.98 | 1343.48 |
| | NV-Embed-v2 (Lee et al., 2025) | 648.96 | 648.96 | 613.24 | 6232.20 | 6232.20 | 6167.62 |
| | HippoRAG 2 (Gutiérrez et al., 2025) | 2896.50 | 2896.50 | 2554.56 | 26407.16 | 26407.16 | 23464.18 |
| | EPIC | **0.45** | **1.54** | **9.70** | **4.84** | **15.48** | **101.30** |

# J LLM USAGE

We used an LLM to refine the sentences and ensure grammatical accuracy.

