# OpenReview forum: "EPIC: Efficient Personalized Index Construction for Retrieval-Augmented Generation"
_ICLR.cc/2026/Conference — ICLR 2026 Conference Withdrawn Submission_

### Official Review · Reviewer_6dF8 · 2025-10-26

**Soundness:** 2
**Presentation:** 2
**Contribution:** 1
**Rating:** 2
**Confidence:** 4

**Summary:**

The paper presents a solution to the scalability and personalization challenges facing Retrieval-Augmented Generation (RAG) in personalized AI assistants. The core contribution is the EPIC framework, which integrates user preferences into both the indexing and retrieval pipelines. This is achieved through two novel components: Preference-Aware Memory Refinement, which uses a coarse-to-fine filtering and rewriting process to prune a massive corpus into a significantly more compact, preference-aligned index (achieving a reported 1110x memory reduction), and Preference-Guided Embedding Steering, which adjusts the query embedding during retrieval to maximize alignment with the user's documented preferences.

The authors successfully demonstrate that EPIC is a highly effective, plug-and-play optimization. The framework substantially boosts both efficiency (drastically cutting memory and latency) and accuracy (improving performance by over 10%p compared to strong baselines like HippoRAG 2) on the task of generating preference-aligned responses. They plan to release three new large-scale preference-aware RAG datasets that could potentially also offer a valuable resource for future research in this domain.

**Strengths:**

1. Preference-Aware Memory Refinement. This process drastically prunes the index, achieving a reported 1110x reduction in memory consumption and a 110x decrease in retrieval latency compared to baselines.
2. The framework successfully integrates user preferences into the entire RAG pipeline and validates its contribution using both performance and operational metrics.
3. The ablation experiments are complete and sound, assessing the contribution of each of the frameworks' components.

**Weaknesses:**

1. Misleading Problem Framing: The paper positions itself as a contribution to "Retrieval-Augmented Generation" (RAG), but the proposed method, EPIC, innovates exclusively on the retrieval pipeline (indexing and retrieval). The experimental setup confirms this by isolating the retriever and keeping the generation component fixed. The work should be more precisely framed as a contribution to personalized retrieval, as it does not rely on the generative component of the RAG pipeline.

2. Incomplete Literature Review and Baselines: The paper tackles two well-studied problems (memory cost and preference-aware retrieval) but overlooks significant bodies of related work.
    a. For memory reduction, the work fails to discuss or compare against prominent, low-cost approaches like vector quantization or binary embeddings, which are standard for reducing memory footprints.
    b. For personalization, the paper does not engage with the extensive literature on preference-based reranking (e.g., UNICORN) or other methods for injecting user preferences at query time, which represent a large class of competing solutions.

3. Unanalyzed Indexing Costs and Factual Integrity Risks: The "Fine-grained filtering & rewriting" stage relies on two LLM-based components (Decision Module and Rewriter). The paper does not adequately analyze the significant computational, financial, and latency costs of making these LLM calls per-document during indexing. More critically, the Rewriter component actively modifies source documents. This introduces a severe risk of factual distortion, omission of key details, or hallucination, yet the paper provides no mechanism for verifying the factual integrity of the rewritten text.

4. Assumption of Static Data and Preferences: The paper motivates the problem by citing "a continuous stream of evolving data," yet the proposed solution (EPIC) constructs a static, pre-filtered index. It is unclear how the system adapts to evolving documents or, more importantly, evolving user preferences.

**Questions:**

1. Were the experiments conducted with a strict split between the preferences used for index creation (i.e., "seen" preferences) and those used for evaluation ("unseen" preferences)? If not, the results may not reflect the system's ability to generalize. If such a split was used, please detail the methodology. How does the system's performance degrade when evaluated on preferences that are substantially different from those used to build the index?

2. The paper introduces "Pref-Appended (Zhou et al., 2024)" as a personalized retrieval baseline. However, the (Zhou et al., 2024) citation is not discussed in the related work section, and the paper itself appears to introduce a method named "CoPS". Could the authors please clarify what specific method was implemented as 'Pref-Appended' and confirm that it is the most appropriate and strongest baseline for this type of query-time preference injection?

---

### Official Review · Reviewer_jemV · 2025-10-31

**Soundness:** 3
**Presentation:** 3
**Contribution:** 4
**Rating:** 6
**Confidence:** 3

**Summary:**

This paper examines the problem of incorporating personal preferences into a RAG system. The goal is to populate a relevant, compact database that aligns with a set of already-populated user preference embeddings. The procedure can be broken down into three stages: (1) a coarse filtering step that relies only on the embedding similarity (2) a LLM-based fine-grained filtering and re-write step based on the textual content of the document (3) a embedding steering step that further aligns the embedding of the data instance with the corresponding user preference vector.

The paper proposes three largely LLM-built datasets to evaluate the usefulness of the proposal. Experimental results show that the proposal was able to achieve better performance and smaller memory footprint compared to static retrievers, as well as LLM-augmented indexing frameworks. Further analyses show that each of those three stages are essential for the final performance gain, and coarse filtering is a very cheap yet effective step to reduce the overall index size and latency.

**Strengths:**

1. The created dataset is going to be very useful for subsequent work in this space.
2. The experimental results clearly demonstrate that the proposed method is effective and efficient (both in terms of indexing and query) with incorporating user preferences.
3. Comparison includes strong baselines such as NVEmbed and HippoRAG, which further strengthens the argument.

**Weaknesses:**

1. One minor concern I have about the credibility of the results is that I can't find much detail about the LLM-as-a-judge setup. The argument would have been more convincing if more details were provided regarding (1) LLM used, (2) the validation of LLM-as-a-judge results. Some human evaluation on a small sample would also greatly strengthen the argument.
2. I struggled with understanding at the beginning because the paper didn't give any concrete definition of "user preference" until Section 4. I also wasn't aware that all user preferences are available in advance (L284) until pretty late in the paper. I think the reader would benefit from a problem definition subsection either in the intro or at the beginning of Section 3.

**Questions:**

1. To confirm my understanding: how did you encode preference in your experiments? Did you make one embedding per entry? (e.g. "I dislike pickup trucks because I find them too large and impractical.")
2. I know all user preferences are pre-populated, but can the authors comment a bit about real-life deployment situation? Would the proposal still apply under a streaming setup of user preferences (imagine it is being updated in a online fashion as the user keeps making more queries)?

---

### Official Review · Reviewer_br1e · 2025-11-01

**Soundness:** 2
**Presentation:** 3
**Contribution:** 2
**Rating:** 2
**Confidence:** 4

**Summary:**

In this paper, the authors introduce EPIC, a framework designed to address two critical challenges in personalized Retrieval-Augmented Generation (RAG): prohibitive memory costs from indexing large corpora and preference misalignment in standard retrievers.  EPIC integrates user preferences into both the indexing and retrieval stages through two core components: (1) preference-aware memory refinement: a two-stage (coarse-to-fine) process that first filters documents using embedding similarity and then uses an LLM for fine-grained verification and rewriting. (2) preference-guided embedding steering, which adjusts query embeddings towards the direction of relevant user preferences at retrieval time, ensuring the retrieval process itself is preference-aware. Three datasets, PrefWiki, PrefRQ, and PrefELIS, are used to evaluate the proposed method.  Experimental results show that EPIC significantly outperforms baseline methods.

**Strengths:**

(1) The problem discussed in the paper is well-motivated and important. The paper identifies two practical bottlenecks for real-world personalized AI assistants: memory scalability and preference alignment.

(2) The proposed method is reasonably designed, with careful descriptions of the components.

(3) Experimental evaluation is comprehensive: The paper provides a thorough evaluation across three new datasets and multiple LLM backends. The comparison against a wide range of baselines is convincing and demonstrates EPIC's superiority across key metrics: accuracy, memory, and latency.

**Weaknesses:**

(1) The problem of the assumption: selection of preference-relevant content for indexing is problematic and risky when user interests are dynamic. Explicit preferences of users are usually unknown. The paper does not address how to infer implicit and dynamic preferences from user behavior or how to handle preference drift over time, which are crucial for long-term personalization. Furthermore, interests are usually unknown when documents are processed. The authors should better justify the assumption, for example, by conducting studies of real user behavior.

(2)  Limited exploration of "cold-start" scenarios: The paper assumes each query is associated with at least one known preference, avoiding the cold-start problem. The framework's performance for new users or queries with no clear preference mapping is not explored.

(3) Efficiency problem: as the authors claim, the corpus is large. While the coarse-grained filtering is fast, the fine-grained stage requires an LLM call per candidate document. Although the coarse stage drastically reduces this number, the cost and latency of this step could still be non-trivial for extremely large corpora or frequent updates. A more detailed discussion on the computational cost of this stage would be beneficial.

**Questions:**

(1) Dynamic preferences: How would EPIC be adapted to handle evolving or dynamic user preferences? Would the entire index need to be reconstructed from scratch, or could an incremental update mechanism be designed?

(2) Robustness to noisy preferences: How robust is the framework to ambiguous or poorly formulated user preferences?

(3) Could you give some real application scenarios or data studies to support the assumption of explicit & static user preferences?

---

### Note · Authors · 2025-11-13

I have read and agree with the venue's withdrawal policy on behalf of myself and my co-authors.